# Diminished Pathogen and Enhanced Endophyte Colonization upon CoInoculation of Endophytic and Pathogenic *Fusarium* Strains

**DOI:** 10.3390/microorganisms8040544

**Published:** 2020-04-09

**Authors:** Maria E. Constantin, Babette V. Vlieger, Frank L. W. Takken, Martijn Rep

**Affiliations:** Molecular Plant Pathology, Faculty of Science, Swammerdam Institute for Life Sciences, University of Amsterdam, 1098 XH Amsterdam, The Netherlands; m.e.constantin@uva.nl (M.E.C.); f.l.w.takken@uva.nl (F.L.W.T.)

**Keywords:** colonization, *Fusarium*, endophyte, Fo47, wilt disease

## Abstract

Root colonization by *Fusarium oxysporum* (Fo) endophytes reduces wilt disease symptoms caused by pathogenic Fo strains. The endophytic strain Fo47, isolated from wilt suppressive soils, reduces *Fusarium* wilt in various crop species such as tomato, flax, and asparagus. How endophyte-mediated resistance (EMR) against *Fusarium* wilt is achieved is unclear. Here, nonpathogenic colonization by Fo47 and pathogenic colonization by Fo f.sp. *lycopersici* (Fol) strains were assessed in tomato roots and stems when inoculated separately or coinoculated. It is shown that Fo47 reduces Fol colonization in stems of both noncultivated and cultivated tomato species. Conversely, Fo47 colonization of coinoculated tomato stems was increased compared to single inoculated plants. Quantitative PCR of fungal colonization of roots (co)inoculated with Fo47 and/or Fol showed that pathogen colonization was drastically reduced when coinoculated with Fo47, compared with single inoculated roots. Endophytic colonization of tomato roots remained unchanged upon coinoculation with Fol. In conclusion, EMR against *Fusarium* wilt is correlated with a reduction of root and stem colonization by the pathogen. In addition, the endophyte may take advantage of the pathogen-induced suppression of plant defences as it colonizes tomato stems more extensively.

## 1. Introduction

Among the most common inhabitants of soils are fungal isolates belonging to the *Fusarium oxysporum* (Fo) species complex [1,2,3]. Their presence is not limited to the soil—*Fusarium* hyphae can also colonize plant roots superficially and internally. The typically asymptomatic root colonization by Fo shows that Fo is mostly an endophyte [2,4]. Endophytic colonization is often restricted to the root cortex and endodermis, and fungal hyphae do not commonly reach xylem vessels. In resistant plants, limited spread of pathogenic isolates in the xylem correlates with the production of gums and tyloses, apparently halting the fungus at the early stages of infection [5]. In susceptible plants, this response appears to be induced too late, and the multiple occlusions that eventually block the xylem vessels affect the water transport of the plant, leading to wilting and death [6].

Pathogenic Fo isolates are currently grouped into 106 *formae speciales* (ff. ssp.) [7]. One of these is Fo f.sp. *lycoperisici* (Fol), which causes wilt disease in tomato plants. Disease symptoms caused by Fol can be strongly reduced upon pre or coinoculation with endophytic Fo strains [8,9]. One of the first *Fusarium* endophytes shown to reduce *Fusarium* wilt disease is Fo47, which was isolated from wilt-suppressive soils [10]. Since its discovery, Fo47 has been shown to reduce *Fusarium* wilt in a variety of plant species including tomato, asparagus, flax, chickpea, and cotton [8,9,11,12,13,14]. Endophyte-mediated resistance (EMR) against *Fusarium* wilts is not unique to Fo47, as other Fo isolates also have this capacity [15]. Moreover, even isolates pathogenic on another host can confer resistance, such as Fo f.sp. *dianthi* against Fol in tomato plants [16]. Additionally, other *Fusarium* species seem to be able to reduce disease symptoms induced by Fo [15,17]. For example, the *Fusarium solani*-K isolate, which colonizes tomato roots, reduces susceptibly against Fo f.sp. *radicis-lycopersici* [17]. These observations suggest that roots colonized by endophytic *Fusarium* are less susceptible to *Fusarium* wilt.

Currently, it remains unclear how *Fusarium*-mediated resistance against *Fusarium* wilt is achieved. A recent report showed that tomato lines deficient in salicylic acid accumulation, jasmonic acid biosynthesis, or ethylene production and sensing could still trigger EMR against *Fusarium* wilt disease [9]. To better understand the extent to which EMR may be host genotype-specific, a range of tomato lines and species were inoculated. Moreover, the contribution of the endophyte genotype in triggering EMR by using various *Fusarium* species was determined. Colonization of tomato roots by endophytic and pathogenic strains was assessed, as was the migration into the stem upon single and coinoculation. Our results showed that various *Fusarium* species can behave as endophytes in tomato roots and trigger similar resistance levels across different tomato species. Roots and stems of tomato plants coinoculated with Fo47 and Fol were found to be colonized to a lesser extent by Fol than when inoculated alone, but surprisingly Fo47 colonization of stems was enhanced in the presence of the pathogen.

## 2. Materials and Methods

### 2.1. Plant Lines and Fungal Strains

Bioassays were performed either on the tomato (*Solanum lycopersicum*) line C32 or Money Maker, susceptible to *Fusarium* wilt, or on the wild tomato relatives *S. pimpinellifolium* (accession LA1578) and *S. chmielewskii* (accessions LA1840, LA2663 and LA2695) in a climate-controlled greenhouse with a day-night temperature of 25°C, 16 h light/ 8 h dark, and a relative humidity of 65%. Fol029 (race 3, sFP2381, carrying phleomycin resistance [18]) and Fo47 (sFP1544, carrying hygromycin resistance, [19]) were inoculated on the aforementioned tomato lines. Bioassays using various *Fusarium* species were performed on the tomato line C32 using the wild-type pathogen Fol4287 (sFP801) [20], and endophyte Fo47 (sFP730) [10], *F. hostae* (sFP2236), *F. proliferatum* (sFP2240), *F. redolens* (sFP4856) [21], and *F. solani* (sFP895). To facilitate fungal reisolation from tomato stems transgenic fungi caring different resistance markers were used. Therefore, the endophytic strain Fo47 containing hygromycin resistance (sFP1544) and the pathogenic strain Fol4287 (race2, SFP3858) [22] or Fol029 (race 3, sFP2381) were used for the reisolation experiments. Strains used in bioassays are described in the supplementary material are the following: Fo47 (sFP1544, [19]), Fol4287 (sFP3059, [23]), Fo47 (sFP730), Fol4287 (sFP801), and Fol017 (sFP17, [24]).

### 2.2. Fusarium Inoculation and Disease Scoring

*Fusarium* strains were grown on potato dextrose agar (PDA) plates for seven to ten days at 25°C in the dark. An agar plug from these plates was transferred to 100 mL of NO3 minimal media (1% KNO3, 0.17% Yeast Nitrogen Base without amino acids and ammonia and 3% sucrose) and incubated at 150 rpm for 3-5 days at 25°C. Spores were filtered over one layer of miracloth filter (Millipore), centrifuged at 2000 rpm, washed with sterile water, and finally resuspended in sterile MiliQ-water. Ten-day-old or 13-days-old tomato seedlings were uprooted, and their roots were trimmed to facilitate *Fusarium* infection. Subsequently, tomato roots were dipped for five minutes in a suspension of 107 spores/mL or 107 spores/mL: 107 spores/mL (ratio 1:1) in the case of the coinoculation treatments if not stated differently. After inoculation, the seedlings were potted, and three weeks afterwards fresh weight and disease index were assessed as described [9]. In short, the disease score was based on the number of brown vessels and external growth wilting symptoms, where 0= healthy plant with no brown vessels; 1= brown vessel(s) only at the basal level; 2= one or two brown vessels at the cotyledon level, the plant still looks healthy; 3= at least three brown vessels and the plant shows clear external witling symptoms; 4= all vessels are brown and the plant shows clear size reduction; and 5= plant is dead. Fresh weight was measured by determining weight of each tomato plant cut at cotyledon level.

### 2.3. Fungal Recovery Assay

Tomato stems were collected three weeks after inoculation and surface sterilized with 70% ethanol as described [9]. In short, the ethanol was removed by pouring into a collection tube, and stems were rinsed twice with sterile water to remove the ethanol. Subsequently, the extremities of the stem were trimmed and one piece at the cotyledon and one at crown level were cut and placed on PDA plates containing 200 mg/L streptomycin and 100 mg/l penicillin to prevent bacterial growth. When transgenic fungal strains were used, the PDA plates also contained 100 mg/L zeocin or 100 mg/L hygromycin for ensuring selection of the right fungal strain. Plates were incubated for four days at 25°C in the dark, after which Fo outgrowth was assessed as follows: 0= no fungal outgrowth, 1= fungal outgrowth of stem pieces isolated from either crown or cotyledon level, and 2= fungal outgrowth from stems isolated at both crown and cotyledon level.

### 2.4. Fungal DNA Isolation and Sequencing

To confirm that the mycelium emerging from tomato stems corresponded to the originally inoculated strain, the mycelium was scraped from the PDA plate and used for gDNA isolation and PCR as described by the authors of [25]. In short, mycelium scrapes were placed in a 2 mL tube containing 400 µL TE buffer (1 mM EDTA pH = 8, 10 mM Tris pH= 8), 200-300 µL glass beads, and 300 µL phenol:chloroform (1:1), followed by disruption using a TissueLyser (Qiagen; Venlo, Netherlands) for 2 min at 30 Hz. Afterwards, the samples were centrifuged for 10 min. The upper phase was transferred to a new 2 mL tube and was subjected to another round of phenol:choloroform extraction. PCR amplification of *EF1-alpha* gene (see Appendix A for primers) was performed in 20 µL reaction consisting of of 0.4 µL dNTPs (10 mM), 4 µL SuperTaq buffer (10x), 0.1 µL SuperTaq (5 U/µl), 1 µL of DNA template, and 1 µL primers (5 pmol/µL). The cycling program for PCR amplification was 95°C for 5 min, 35 cycles of 30s at 95°C, 30s at 55°C, and 1 min at 72°C, with a final elongation step of 3 min at 72°C. The PCR amplicon was sequenced and compared with the *EF1-alpha* sequence of the strain inoculated originally.

### 2.5. Analysis of Fungal Colonization by Quantitative PCR

Tomato roots were harvested three weeks after inoculation, washed, snap-frozen in liquid nitrogen, and freeze-dried overnight. Samples were ground in a mortar cooled with liquid nitrogen using a pestle and approximately 100 mg of the resulting powder was using for gDNA isolation. gDNA isolation and purification were performed using GeneJET plant Genomic purification Kit (Thermo Scientific; Walthamm MA, USA). DNA concentration was estimated by spectroscopy using Nanodrop (Thermo Scientific, Walthamm MA, USA), and the quality of the gDNA was assessed by agarose gel electrophoresis. The 10 µL qPCR mixture contained 10 ng of gDNA, 10 pM of each primer, and 2 µL of HOT FirePolEvaGreen qPCR Mix Plus (Solis BioDyne; Tartu, Estonia) were performed in QuantStudioTM3 (Thermo Scientific; Walthamm MA, USA). The cycling program was set to 15 min at 95°C, 40 cycles of 15s at 95°C, 1 min at 60°C, and 30s at 72°C. The melting curve analysis was performed afterwards as follows: 15s at 95°C, 1 min at 60°C, and 15s at 95°C. The sequences of the primers used are summarized in Appendix A [26,27]. For InterGenic Spacer (IGS) primers two standard curves (four- or ten- times dilution) were performed with a starting concentration of 10 ng, resulting with a primer efficiently of 110 and 95%, respectively. Three technical replicates were used per biological sample, and data was normalized to plant tubulin gene level, using qbase+3.1 (Biogazelle; Ghent, Belgium).

### 2.6. Statistical Analyses

Data collected from bioassays (fresh weight, disease index) were analyzed using PRISM 7.0 (GraphPad) by performing a Mann–Whitney U- test. The data obtained from qPCR were analyzed with ordinary one-way ANOVA with Tukey’s multiple comparisons test in the case of IGS and with an unpaired Student’s t-test for *SIX8* and *SCAR*.

## 3. Results

### 3.1. Endophyte-Mediated Resistance Occurred at a 1:1 Ratio and Required Live Endophyte Spores

The endophytic strain Fo47 has been reported to trigger EMR when inoculated in plants at a concentration 10-100 times higher than that of the pathogen [28]. To determine the optimal concentration for a reliable EMR assay, tomato plants were coinoculated with Fo47 at the same ratio as the pathogen (1:1), or in 10- or 100-times excess (Appendix A). Decreasing the pathogen concentration lowered the severity of disease symptoms (Appendix A) but did not influence the level of EMR triggered by Fo47, since it was already quite strong at a 1:1 endophyte: pathogen ratio (Appendix A). Based on this, all subsequent bioassays were carried out using 10^7^ spores/mL for single inoculations and ratio 1:1 for coinoculation treatment. To determine whether a living endophyte is required to induce resistance, tomato seedlings were coinoculated with heat-killed spores of either Fo47 or Fol4287, together with living Fol4287 (Appendix A). Since plants coinoculated with heat-killed spores and Fol4287 became diseased, it can be concluded that a living endophyte is required for triggering EMR (Appendix A).

Fo strains within the same vegetative compatibility group (VCG) can form stable heterokaryons, while those belonging to different VCGs undergo cell death upon heterokaryon formation [29]. To test whether VCG incompatibility can result in a reduced viable concentration of Fol4287, and thereby reduce disease symptoms, two Fol strains of different VCG (Fol4287 (VCG030) and Fol017 (VCG031), [26]) were coinoculated. This resulted in disease symptoms that were indistinguishable from those observed upon single inoculation (Appendix A). This observation implies that VCG incompatibility is unlikely to be an explanation for the suppression of disease symptoms by Fo47. Finally, to determine whether there is direct antagonism between Fo47 and Fol4287, two agar plugs with seven days old mycelium were placed on a PDA plate at a fixed distance from of each other. No visible inhibition halo or reduction of mycelial growth was observed (Appendix A). Taken together, living Fo47 spores can efficiently confer resistance against Fol4287 when applied in a 1:1 ratio, without exhibiting obvious direct antagonism.

### 3.2. Fo47 Also Conferred Resistance against Fol in Wild Tomato Species

To test whether Fo47 can also trigger EMR in wild tomato species, 13-days-old uprooted tomato seedlings were inoculated with either water (mock), a spore solution of either Fo47 or Fol029, or both Fo47 and Fol029 in a 1:1 ratio. Based on the severity of disease symptoms, each plant received a disease index (DI) score ranging from 0 (healthy) to 5 (dead). Fo47 treatment caused no visible disease symptoms in any plant line, and fresh weight was indistinguishable from the mock (Figure 1b,c). Conversely, inoculation with Fol029 caused a visible growth reduction in all plant lines (Figure 1a). The cultivated tomato line *S. lycopersicum* C32 showed the most drastic weight reduction among the three-tomato species tested (Figure 1a,b) and displayed the most consistent disease symptoms (Figure 1c). *S. pimpinellifolium* and *S. chmielewskii* exhibited more varying disease symptoms (DI = 2 to 5) (Figure 1b and Appendix Aa,b) upon Fol029 inoculation. Coinoculation of Fo47 with Fol029 resulted in increased fresh weight compared with solely Fol029 inoculated tomato lines (Figure 1a,b), but this difference was only statistically significant for *S. lycopersicum* and *S. chmielewskii* line LA2663. Similarly, tomato lines coinoculated with Fo47 and Fol029 showed reduced disease symptoms compared with plants single inoculated with Fol029 (Figure 1c, Appendix Aa,b). In conclusion, Fo47 reduced disease symptoms caused by Fol029 in both cultivated and wild tomato species.

### 3.3. Different Fusarium Species Behaved as Endophytes in Tomato Plants and Triggered Resistance against Fol

To test if the ability of conferring EMR against *Fusarium* wilt is more broadly present in the genus *Fusarium* tomato, bioassays with *Fusarium* spp. other than Fo were performed. Roots of tomato seedlings were inoculated in water or spore suspension of Fo47, *Fusarium redolens* (Fr), *Fusarium solani* (Fs), *Fusarium hostae* (Fh), or *Fusarium proliferatum* (Fp), together with Fol4287 in a 1:1 ratio. Inoculation of Fo47 or other *Fusarium* spp. alone did not result in visible differences compared to the mock inoculation control (Figure 2a) or in differences in fresh weight (Figure 2b). Therefore, the Fr, Fs, Fh, and Fp strains used here are not pathogenic on tomato plants (Figure 2c). Tomato plants inoculated with Fol4287 showed a reduction in fresh weight (Figure 2b) compared to the mock treatment and developed disease symptoms three weeks after inoculation (Figure 2c). In this experiment, these disease symptoms were not as severe as previously observed (Appendix A). Tomato plants coinoculated with Fol4287 and other *Fusarium* isolates (such as Fo47, Fr, Fs, Fh, and Fp) were taller than plants inoculated with Fol4287 alone (Figure 2a). Differences in fresh weight were significant for coinoculation with Fo47, Fr, and Fs (Figure 2b). In line with this, coinoculation treatment of Fol4287 with a nonpathogenic *Fusarium* strain resulted in reduced disease symptoms compared with Fol4287 alone (Figure 2c, Appendix Ac). Fh and Fp did not consistently reduce *Fusarium* disease symptoms, suggesting that they may have a weaker effect than the other isolates tested.

To examine whether the different *Fusarium* spp. are endophytes, tomato stems were harvested three weeks post inoculation, surface sterilized, and a piece of the stem at crown level was harvested. These were placed on PDA plates with antibiotics and incubated at 25°C as schematically depicted in Appendix A. After four days, mycelia emerged from stem pieces, which was used for gDNA isolation and *EF1-alpha* PCR followed by sequencing of the PCR product. This analysis revealed that the mycelium originated from the *Fusarium* strain used to inoculate the plants (Figure 2d). Following mock treatment, either no fungal mycelia emerged from the stems (Figure 2d), or mycelia emerged that did not correspond to *Fusarium* spp. (Appendix A). The most frequently reisolated fungus from tomato stems was Fol4287, followed by Fo47, Fp and Fr, and Fs (Figure 2d, Appendix A). The least frequently reisolated species from tomato stems was Fs (Figure 2, Appendix A); however, its presence was confirmed in one tomato stem. Taken together, Fr, Fs, Fh, and Fp can behave as endophytes, colonizing tomato stems and triggering EMR against Fo wilt in tomato.

### 3.4. Coinoculation of Fol with Fo47 Limited Colonization of tomato stems by Fol4287 while Fo47 Colonization was Increased

To determine the extent of tomato stem colonization of Fo47 and Fol4287 upon coinoculation, these strains were reisolated from tomato stems three weeks after inoculation. To discriminate between the strains, transgenic fungi carrying either hygromycin (Fo47) or phleomycin (Fol4287, Fol029) resistance were used in this experiment. As reported [9], Fo47 colonization was usually restricted to the crown level, and the fungus was rarely observed at cotyledon level (Figure 3a,b). In contrast to Fo47, Fol4287 was reisolated from both crown and cotyledon level in every case upon single inoculation (Figure 3a,b). Coinoculation of Fo47 and Fol4287 strongly reduced the extent of colonization by Fol4287 in tomato stems, and in few instances Fol4287 could not be reisolated from either crown or cotyledon level (Figure 3a,b). Surprisingly, coinoculation of Fo47 and Fol4287 led to more frequent reisolation of Fo47 from tomato stems at crown level and even cotyledon level (Figure 3a,b). This reduced extent of stem colonization by the pathogen and increased migration into the stem by Fo47 upon coinoculation was also observed in wild tomato species (Appendix A). It appears therefore that Fo47 can limit the spread of the pathogen in stems of various tomato species, while Fo47 colonization is enhanced in tomato stems when coinoculated with a pathogenic strain.

### 3.5. Fo47 Limited Fol Colonization in Tomato Roots

Next, the level of root colonization by Fo47 or Fol4287 was determined to assess whether it changes upon coinoculation. To do so, fungal biomass in tomato roots was measured by quantitative PCR (qPCR). Tomato roots were harvested three weeks after inoculation, and qPCR was performed using either Fo primers designed for InterGenic Spacer (IGS) region or primers for Fo47 (*SCAR* primers described [27]) and for Fol4287 (SIX8 primers) relataive to the plan tubulin. In line with the stem reisolation experiment, Fol4287 was found to colonize tomato roots approximately 20-fold more than Fo47 (Figure 4a). Upon coinoculation, total fungal biomass in tomato roots was similar to tomato roots only inoculated with Fo47 but reduced compared with roots inoculated only with Fol4287 (p = 0.0349, Figure 4a). To distinguish between Fo47 and Fol4287 colonization in coinoculated roots, Fo47- and Fol4287-specific primers were used (Figure 4b,c). Quantification by qPCR revealed that Fo47 colonization of tomato roots is similar in single inoculated or coinoculated roots (Figure 4b). In contrast, Fol4287 colonization is about 9-fold reduced in tomato roots coinoculated with Fo47 compared to when inoculated alone (Figure 4c). Overall, our data show that Fo47 is a poor root colonizer compared with Fol4287, but can drastically reduce the amount of Fol4287 in tomato roots upon coinoculation.

## 4. Discussion

Fo47 triggered resistance against Fol029 in two wild tomato species to a similar extent as in cultivated tomato. Various studies have reported that Fo47 can trigger resistance in a variety of plants: asparagus, flax, cucumber, eucalyptus, pepper, banana, and chickpeas [30]. This suggests that EMR is a conserved host trait. Coinoculation of Fo47 with Fol4287 reduced proliferation of the pathogen in both stems and roots, while colonization by the endophyte was enhanced only in stems. Therefore, EMR is likely achieved by limiting pathogen colonization of tomato roots and stems.

Additionally, like Fo, *F. redolens*, *F. solani*, *F. hostae*, and *F. proliferatum* behaved as endophytes and conferred protection against *Fusarium* wilt disease in tomato plants. Previous studies in tomato plants found that the endophytic strain *Fusarium solani*-K could confer resistance against Fo f.sp. *radicis-lycopersici* (Forl) [17]. Moreover, 205 Fo strains, 81 *F. proliferatum* strains, and other 16 *Fusarium* spp. have been shown to confer resistance against Forl [15]. This suggests that many *Fusarium* spp. share the capability of being endophytes and triggering EMR against *Fusarium* wilt.

Fungal quantification of infected tomato roots revealed that Fo47 drastically reduces root colonization by Fol4287. This finding is in line with an earlier study in which the biomass of pathogenic strain Fol8 was significantly reduced in tomato roots in presence of Fo47 [8]. Additionally, a reduction of Fol4287 and Fol029 colonization of tomato stems when coinoculated with Fo47 was observed. The lower amount of pathogen in roots and stems upon coinoculation correlates with the reduced disease symptoms observed. Similarly, the endophytic *Verticillium* strain Vt305 was observed to reduce the amount of the pathogenic species *Verticillium longisporum* in cauliflower roots, hypocotyl, and stem. This suggests that endophytes can trigger resistance by reducing the proliferation of a pathogen in the plant [31].

Strain Fo47 colonized tomato stems to a greater extent when coinoculated with Fol4287, compared to single inoculation, despite not colonizing the tomato roots more extensively. One possible explanation for this phenomenon is that Fo47 takes advantage of effector-triggered susceptibility facilitated by Fol. Fol is known to secrete effector proteins, such as Avr2, which facilitate Fol colonization of tomato stems [32]. Transgenic tomato stems over-expressing AVR2 without signal peptide are hyper-colonized by Fo47, compared to wild-type plants [33]. Therefore, it seems plausible that Fo47 benefits from Fol effector-triggered immune suppression to more extensively colonize tomato stems. There are other examples of such a “boost” in endophytic stem colonization facilitated by pathogens. The *Verticillium* endophyte Vt305 was found to accumulate to higher levels at 70 dpi in tomato stems but not the roots when coinoculated in a 1:1 ratio with the pathogen *Verticillium longisporum* [31]. In another case, infection of *Arabidopsis* plants with the white rust pathogen *Albugo* facilitated infection by *Phytophthora infestans*, which otherwise does not infect *Arabidopsis* [34].

Several questions about EMR remain unanswered. One is how Fo47 reduces root colonization by Fol. One possibility is that Fo47 limits the spread of Fol via direct antagonism, such as antibiosis or competition. Another possibility is indirect antagonism by inducing host resistance. These two possibilities are not mutually exclusive. No antibiosis was observed under in vitro conditions, and in our bioassay set-up it is unlikely that there is competition for growth on roots or for infection sites, since *Fusarium* was inoculated by exposing damaged roots to spores before transplanting them to the soil. Still, in planta competition between strains cannot be excluded, for instance, Fo47 might consume nutrients limiting Fol development. This hypothesis is difficult to test, and it does not easily fit with the observation that endophytic Fo can trigger resistance in a split root system against pathogenic Fo [13,16]. Additionally, the low colonization level of Fo47 in tomato roots seems unlikely to have a drastic impact on the level of nutrients. It is more likely, therefore, that Fo47 induces resistance responses in the host that result in reduced Fol colonization of roots and subsequent spread in the stems. The underlying mechanism of this is elusive but appears to be independent of the major defense-related hormones ethylene, salicylic acid, and jasmonic acid. Whether physical barriers such as papillae or lignified cell walls, induced by Fo47 colonization, are involved remains a question for future study [9].

In conclusion, *Fusarium*-mediated resistance against *Fusarium* wilt disease of tomato is common within the genus *Fusarium* and consists of limiting pathogen colonization inside roots and stem, while the extent of endophytic colonization in tomato stems is increased. Understanding how endophytic *Fusarium* species can suppress *Fusarium* wilt disease may help to improve the design of strategies to better control this soil-borne disease.

## Figures and Tables

**Figure 1 microorganisms-08-00544-f001:**
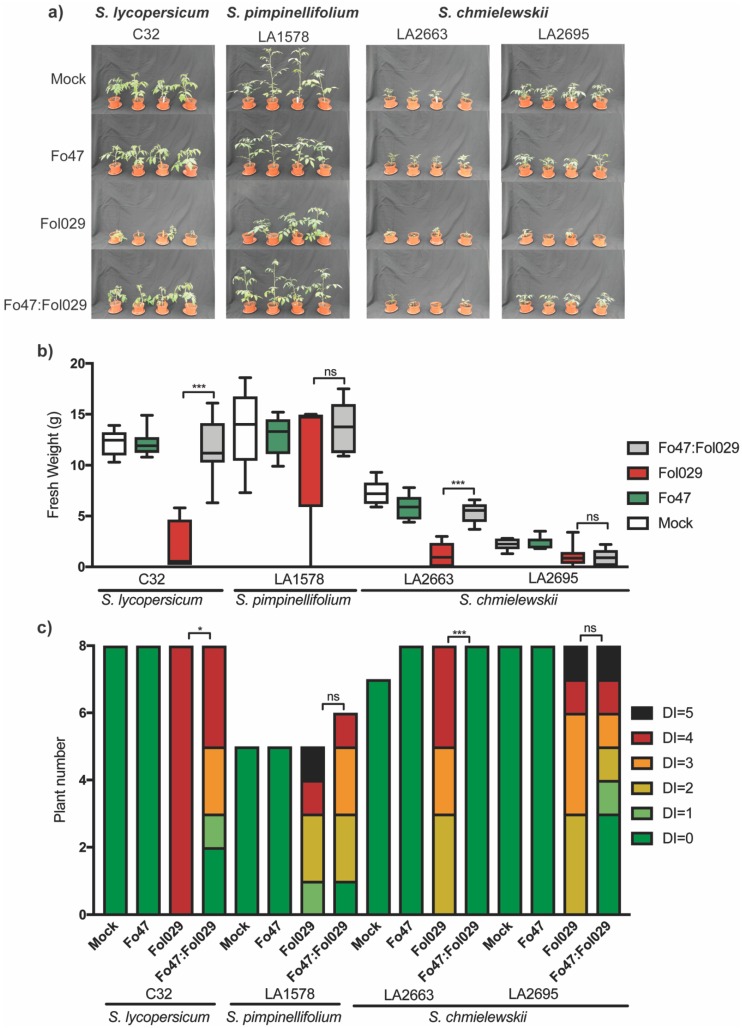
Fo47 can trigger resistance against Fol029 in (**a**) *Solanum lycopersicum* (C-32) and *S. chmielewski* (LA2663, LA2695). Thirteen-day-old tomato seedlings were inoculated with water (mock), Fo47, or Fol029, or coinoculated with Fo47 and Fol029. (**b**) Fresh weight and (**c**) disease index (DI) were assessed three weeks after inoculation where DI = 0 no brown vessels; DI = 1 brown vessel(s) only at basal level; DI = 2 one or two brown vessels at cotyledon level; DI = 3 three brown vessels at cotyledon level; DI = 4 all vessels are brown; DI = 5 the plant is dead. Data were analysed by a nonparametric Mann--Whitney U-test (^ns^ P > 0.05; * *p* < 0.05; *** *p* < 0.001).

**Figure 2 microorganisms-08-00544-f002:**
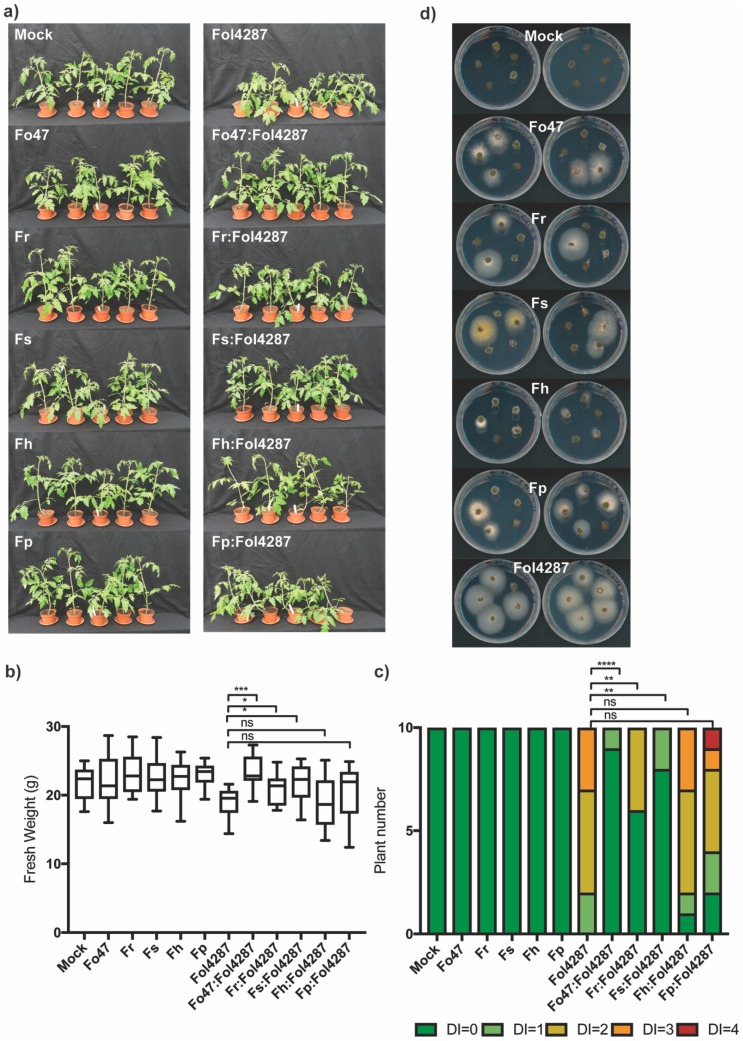
Fo47, *Fusarium redolens* (Fr), *Fusarium solani* (Fs), *Fusarium hostae* (Fh), and *Fusarium proliferatum* (Fp) can reduce *Fusarium* wilt disease symptoms in tomato. (**a**) Representative tomato plants three weeks after inoculation; (**b**) fresh weight and (**c**) disease index (DI) of tomato plants three weeks after inoculation. DI = 0 no brown vessels; DI = 1 brown vessel(s) only at basal level; DI = 2 one or two brown vessels at cotyledon level; DI = 3 three brown vessels at cotyledon level, DI = 4 all vessels are brown, DI = 5 the plant is dead. Data were analysed by a nonparametric Mann--Whitney U-test (^ns^ P > 0.05; * *p* < 0.05, ** *p* < 0.01; *** *p* < 0.001); (**d**) Ten tomato stems pieces from crown level showing *Fusarium* outgrowth on PDA plates after being incubated for four days in dark at 25 °C.

**Figure 3 microorganisms-08-00544-f003:**
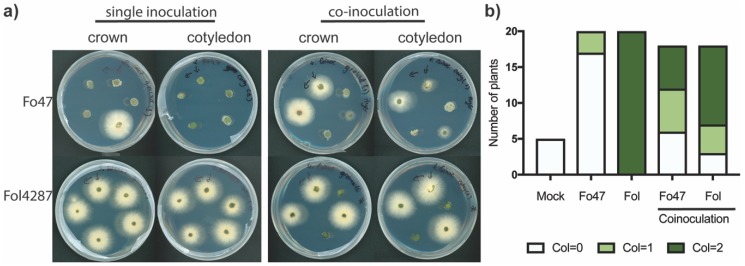
Fo47 colonizes tomato cotyledons upon coinoculation with Fol4287, while Fol4287 colonization is reduced. (**a**) Representative potato dextrose agar (PDA) plates with tomato pieces taken from crown and cotyledon level four days after incubation in the dark at 25°C; (**b**) plates were scored as following: No Fo outgrowth is represented as white (Col = 0), Fo outgrowth at either crown or cotyledon level is represented in light green (Col = 1) an outgrowth at both crown, and cotyledon level (Col = 2) is represented in dark green. This experiment was performed three times with similar results.

**Figure 4 microorganisms-08-00544-f004:**
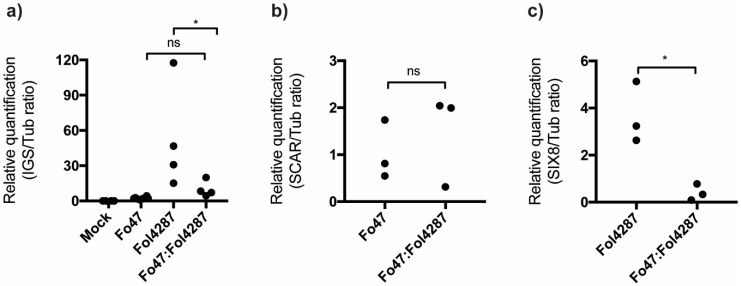
Coinoculated tomato roots are less extensively colonized by pathogenic strain Fol4287. Colonization was assessed by the qPCR using (**a**) Fo InterGenic Spacer (IGS) primers, *(***b**) Fo47 marker *SCAR,* or (**c**) Fol marker *SIX8*. Data obtained by quantifying IGS were analysed by an ANOVA with a Tukey multiple comparison test, and in the case of data for *SCAR* and *SIX8* with a Student’s t-test. (^ns^
*P* > 0.05; * *p* < 0.05; ** *p* < 0.01).

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
