# Peer review of "Diminished Pathogen and Enhanced Endophyte Colonization upon CoInoculation of Endophytic and Pathogenic Fusarium Strains"

_microorganisms, 2020, doi:10.3390/microorganisms8040544_

Round 1

Reviewer 1 Report

Dear Authors,

the article titled "Co-inoculation of endophytic and pathogenic Fusarium strains leads to diminished pathogen and enhanced endophyte colonization" provided a lot of useful information about influence of endophytic fungi on pathogen colonization in various tomato lines. The article is well-written and deserve publication after some improvements.

Point 1: In this work Authors did not use italics for latin names of fungi, which I marked in yellow.

Point 2: Line 66 : "Bioassays were performed either on the susceptible tomato .." please add for what disease this tomato is susceptible, is it Fusarium wilt?

Pont 3: Table S1. Please add the references and annealing temperature for all primers, which you used in this work.

Point 4: Line 88: In chapter Materials and Methods (2.2) you mentioned only that "tomato roots were dipped for five minutes in a suspension of 107 spores/ml or 107 spores/ml: 107 spores/ml (ratio 1:1), however in chapter Results (3.1) you mentioned also about other concentrations of spores, which you used in this experiment. Please explain this matter and eventually provide additional information in Materials and Methods chapter.

Point 5: Please explain what DI abbreviation  mean, placed in Figures 1, S1 or S2 and provide this explanation in material and methods section and particular figures.

Point 6: Please describe strain Fol017 in Materials an Methods, explain why you did not co-inoculate Fo47 and Fol017 the tomato seedlings and did not check the interactions between Fo47 and Fol017 on PDA plates? I think that this result would be very interesting for readers. In Figure S1 and chapter 3.1 could you write which line of tomato did you use.

Point 7: Why you changed the investigated fungal strain in chapter 3.2 and Figure 1? In chapter 3.1 you focused on pathogenic Fol017 and Fol4287 and in chapter 3.2 you focused on Fol029, please explain why.

Point 8: In my opinion in Figure 2c, Fh and Fp can't suppress Fusarium wilt disease in tomato, since those differences are not statistically signifficant, however in Figure S3c have been shown that Fh and Fp can suppress Fusarium wilt disease in tomato, please explain the differences between those two figures and change the Figure 2 title to state the true results according to statistical significance level.

Point 9: Please explain why you described only interactions between Fo47 and Fol4287 and not between Fol029 and Fo47 in the chapter 3.4? Why you add the results for Fo29 only to Figure S5 and not to Figure 3a and 3b? Why you investigated different numbers of plants?

Point 10: Please change the Figure 1 title to state the true results according to statistical significance level, since for the La1578 and La2695 the differences are not significant, so for those lines Fo47 do not trigger resistance against Fol029.

Author Response

Point 1: In this work Authors did not use italics for latin names of fungi, which I marked in yellow.

- has been addressed by formatting in italics all names in Latin.

Point 2: Line 66 : "Bioassays were performed either on the susceptible tomato .." please add for what disease this tomato is susceptible, is it Fusarium wilt?

Addressed by adding “Fusarium wilt”

Pont 3: Table S1. Please add the references and annealing temperature for all primers, which you used in this work.

An additional column entitled annealing temperature was added to the Table S1. The reference style was changed to match the document.

Point 4: Line 88: In chapter Materials and Methods (2.2) you mentioned only that "tomato roots were dipped for five minutes in a suspension of 107 spores/ml or 107 spores/ml: 107 spores/ml (ratio 1:1), however in chapter Results (3.1) you mentioned also about other concentrations of spores, which you used in this experiment. Please explain this matter and eventually provide additional information in Materials and Methods chapter.

Lines 88-89 in the material and methods section were adjusted to clarify the concentrations used. Additional lines concerning the final concentrations used were added in section 3.1.

Point 5: Please explain what DI abbreviation  mean, placed in Figures 1, S1 or S2 and provide this explanation in material and methods section and particular figures.

Figure 1, 2 and 3 were adjusted, as well as Figure S1 ,S2 and S3 to include the disease index (DI) score. Additionally, DI score was explained in the material and methods.

Point 6: Please describe strain Fol017 in Materials an Methods, explain why you did not co-inoculate Fo47 and Fol017 the tomato seedlings and did not check the interactions between Fo47 and Fol017 on PDA plates? I think that this result would be very interesting for readers. In Figure S1 and chapter 3.1 could you write which line of tomato did you use.

Additional information on Fol017 has been added in the material and methods section. Co-inoculation of Fol017 and Fo47 was not tested because we assumed that Fo47 will still confer resistance against this strain. This assumption is based on other experiments, where we could see that Fo47 confers resistance against other Fol strains such as Fol004, Fol007, Fol4287 and Fol029. The lack of negative interactions between Fo47 and Fol4287 on PDA plates was likewise taken as exemplary for any pair of strains, as routinely observed.

Point 7: Why you changed the investigated fungal strain in chapter 3.2 and Figure 1? In chapter 3.1 you focused on pathogenic Fol017 and Fol4287 and in chapter 3.2 you focused on Fol029, please explain why.

Chapter 3.1. was focused on Fol4287, since it a well-studied reference strain. However, for the wild tomato lines race 3 strains such as Fol029 were tested first, because race 3 strains have the highest probability of causing disease and therefore provide an interaction to test EMR. Since, seeds of wild tomato species were limited, we decided to use Fol029, since we know that Fo47 can confer protection against Fol029, but we did not know if these tomato lines would be susceptible to Fol4287.

Point 8: In my opinion in Figure 2c, Fh and Fp can't suppress Fusarium wilt disease in tomato, since those differences are not statistically signifficant, however in Figure S3c have been shown that Fh and Fp can suppress Fusarium wilt disease in tomato, please explain the differences between those two figures and change the Figure 2 title to state the true results according to statistical significance level.

The reviewer in right to that Fh and Fp were not found to significantly suppress disease symptoms by Fol4287 in one experiment (Figure 2) but they do in the second (Figure S3). Since we already noticed this difference, we chose to show the results for both experiments. In the experiment shown in Figure S3, Fol infected plants look relatively healthy, while the experiment shown in Figure 2, the plants shown more visible disease symptoms. Since Fh and Fp show the same trend in reducing disease symptoms upon Fol4287 inoculation we argue that they can suppress, however, these strains may be less consistent suppressors. Section 3.3 was therefore adjusted.

Point 9: Please explain why you described only interactions between Fo47 and Fol4287 and not between Fol029 and Fo47 in the chapter 3.4? Why you add the results for Fo29 only to Figure S5 and not to Figure 3a and 3b? Why you investigated different numbers of plants?

In order to build on previous observations, we generally used Fol4287 for bioassays. However, as explained under point 7, in few cases this was not possible. We chose to show Figure 3 as a main figure since that experiment was performed three times using Fol4287. We chose to leave the other colonization experiment as Figure S5, because it shows combined data of two experiments. Unfortunately, we could not perform the experiments with more plants due to the scarcity of seeds.

Point 10: Please change the Figure 1 title to state the true results according to statistical significance level, since for the La1578 and La2695 the differences are not significant, so for those lines Fo47 do not trigger resistance against Fol029.

The reviewer is right to point out that when analysing disease index for each experiment for line LA1578 and LA2695, no statistical difference was found between Fol029 and co-inoculation of Fo47 with Fol029. However, for line LA2695, when the data of disease index of both experiments are combined, this is statistically significant (as well as for one of the experimental repetitions in Figure S2).

For line LA 1578, both experiments show no statistical difference, despite showing the same trend as the other lines. This could be attributed to the low number of biological replicates used for these experiments . Therefore, we adjusted the text concerning line LA1587 to reflect this. 

Reviewer 2 Report

This interesting paper describes the co-inoculation of non-pathogenic and pathogenic Fusarium strains to reduce plant pathogens. However, the manuscript requires some technical corrections. First of all, the use of the present tense should be avoided in the manuscript. Then, in the chapter Material and Methods, everything assays whose results will be displayed in the chapter Results should be described. The Latin names of fungi write in italic. Likewise, the Latin name of the fungal genus, when writes beside spp., species, genus, and strains, put in also in italic. 

Lines 2-4, write a more concise title

Line 3,  Fusarium strains – Fusarium put in italic

Line 12, instead isolates write strains

Lines 14-17, avoid personal pronouns and the present tense, use past tense for finished results, please change that sentences

Line 15, add „non-pathogenic“ after colonization by; add „strains“ after (Fol)

Line 25, reduce keywords and write a more concise, add „non-pathogenic strain“ in the front of Fo47, the „wild tomato“ put in the end

Line 29,  Fusarium hyphae – Fusarium put in italic

Line 27, in the chapter Introduction, all cited sentences, where authors have already reported results, must be in the past tense, please change it

Line 40,  Fusarium endophytes – Fusarium put in italic

Line 46,  Fusarium species – Fusarium put in italic

Line 48, endophytic Fusarium change in Fusarium endophytes – Fusarium put in italic

Line 50, Fusarium-mediated – Fusarium put in italic

Line 55,  Fusarium species – Fusarium put in italic

Line 57,  Fusarium species – Fusarium put in italic

Line 71,  Fusarium species – Fusarium put in italic

Line 81,  Fusarium strains – Fusarium put in italic

Line 87,  Fusarium infection – Fusarium put in italic

Line 88,  107 spores/ml: 107 spores/ml – 107 spores/ml:107 spores/ml

Lines 126 and 127, Thermo Fisher Scientific instead Thermo Scientific

Line 139, what is fresh weight – seedlings, stems, roots or what, add it

Line 144, change the titles of subchapters in the chapter Results, avoid present tense, please

Line 146, The endophytic strain Fo47

Lines 157-168, presented the results of VCG method, but in the chapter Materials and Methods, description of VCG method is missing

Line 177, abbreviation DI request explanation

Line 186, genus Fusarium, Fusarium spp. – Fusarium put in italic

Lines 187 and 188, names of fungus in italic, example: Fusarium redolens etc.

Line 189, Fusarium spp. – Fusarium put in italic

Line 195, Fusarium isolate put in Fusarium strains (Fusarium put in italic)

Line 196, Fusarium strain – Fusarium put in italic

Lines 206-217, please, Fusarium put in italic

Line 269, deleate (reviewed by

Line 274, avoid personal pronouns, in particular in the present tense

Lines 277 and 278, Fusarium spp. – Fusarium put in italic

Line 291, Verticillium strain – Verticillium put in italic

Line 295, The strain Fo47

Line 303, Verticillium endophyte – Verticillium put in italic

Line 324, deleate „we show that“; Fusarium-mediated – Fusarium put in italic

Line 325, genus Fusarium – Fusarium put in italic

Line 327, Fusarium species – Fusarium put in italic

In the chapter References, please put the regular names of fungi in italic, example: line 369, Fusarium oxysporum instead Fusarium Oxysporum

Author Response

All recommendations (from line 2 to line 327) have been implemented as suggested, with two exceptions:

  • Regarding suggestions concerning line 139, the fresh weight was added at the end of section 2.3. instead of line 139, since we think it be a better fit.

  • Regarding suggestions concerning line 157-168, we did not perform VCG testing, but we used two isolates which were shown previously to belong to different VCGs. Since this was unclear in the text, we adjusted section 3.1.

All references have been corrected.

Round 2

Reviewer 1 Report

Dear Authors,

thank you for the comprehensive answers and corrections.